# Effect of Berberine against Cognitive Deficits in Rat Model of Thioacetamide-Induced Liver Cirrhosis and Hepatic Encephalopathy (Behavioral, Biochemical, Molecular and Histological Evaluations)

**DOI:** 10.3390/brainsci13060944

**Published:** 2023-06-12

**Authors:** Somayeh Hajipour, Yaghoob Farbood, Mahin Dianat, Ali Nesari, Alireza Sarkaki

**Affiliations:** 1Persian Gulf Physiology Research Center, Medical Basic Sciences Research Institute, Ahvaz Jundishapur University of Medical Sciences, Ahvaz P.O. Box 61355-15795, Iran; hajipour.s1984@gmail.com (S.H.); farbood_y@yahoo.com (Y.F.); dianat.m@gmail.com (M.D.); nesari67@yahoo.com (A.N.); 2Department of Physiology, Medicine Faculty, Ahvaz Jundishapur University of Medical Sciences, Ahvaz P.O. Box 61355-15795, Iran; 3Medicinal Plants Research Center, Ahvaz Jundishapur University of Medical Sciences, Ahvaz P.O. Box 61355-15795, Iran; 4National Institute for Medical Research Development “NIMAD”, Tehran 1419693111, Iran

**Keywords:** thioacetamide, hepatic encephalopathy, berberine, oxidative stress, inflammation

## Abstract

Background: Liver cirrhosis (LC) is one of the chronic liver diseases with high disability and mortality accompanying hepatic encephalopathy (HE) followed by cognitive dysfunctions. In this work, the effect of berberine (Ber) on spatial cognition was studied in a rat model of LC induced by thioacetamide (TAA). Materials and Methods: Male Wistar rats (200–250 g) were divided into six groups: (1) control; (2) TAA, 200 mg/kg/day, i.p.; (3–5) TAA + Ber; received Ber (10, 30, and 60 mg/kg, i.p., daily after last TAA injection); (6) Dizocilpine (MK-801) + TAA, received MK-801 (2 mg/kg/day, i.p.) 30 m before TAA injection. The spatial memory, BBB permeability, brain edema, liver enzymes, urea, serum and brain total bilirubin, oxidative stress and cytokine markers in the hippocampus were measured. Furthermore, a histological examination of the hippocampus was carried out. Results: The BBB permeability, brain edema, liver enzymes, urea, total bilirubin levels in serum and hippocampal MDA and TNF-α increased significantly after TAA injection (*p* < 0.001); the spatial memory was impaired (*p* < 0.001), and hippocampal IL-10 decreased (*p* < 0.001). Ber reversed all the above parameters significantly (*p* < 0.05, *p* < 0.01 and *p* < 0.001). MK-801 prevented the development of LC via TAA (*p* < 0.001). Conclusion: Results showed that Ber improves spatial learning and memory in TAA-induced LC by improving the BBB function, oxidative stress and neuroinflammation. Ber might be a promising therapeutic agent for cognitive improvement in LC.

## 1. Introduction

Liver disorders are very frequent and widely distributed in the world. Among them, hepatic fibrosis represents the response of the liver to different chronic insults, and it is accompanied by significant morbidity and mortality [1]. Hepatotoxic agents such as thioacetamide (TAA), carbon tetrachloride (CCl4) and ethionine have been used to initiate cirrhosis in several investigational models. TAA has been widely used for experimental liver injury. While acute TAA application leads to hepatitis, chronic TAA administration has been shown to cause liver cirrhosis [2]. It has been reported that oxidative stress plays a crucial role in the pathogenesis of TAA-induced hepatitis and cirrhosis. Indeed, several lines of research have demonstrated the beneficial effect of antioxidants in protecting the liver against TAA-induced damage [3]. The purpose of the present study was to evaluate the hepatoprotective and cognitive-improving effects of a new drug, berberine, as an antioxidant and anti-inflammatory natural agent.

Acute liver failure caused by toxic liver injury evoked by toxins can lead to central nervous system complications such as hepatic encephalopathy (HE) [4]. Previous investigations identified that bile acid-mediated signaling occurs in the brain during HE and contributes to cognitive impairment [5]. HE is a serious neurological complication of acute and chronic liver failure [4,6]. It is a neuropsychiatric syndrome that develops in subjects with severe liver diseases, with portosystemic shunt surgery in the form of diffuse mild brain edema [7]. HE occurs due to deterioration of hepatic function by liver toxins such as TAA and CCl4 or bile duct ligation. An experimental rat model of HE is induced by the administration of TAA. This model is very comparable to human samples that have progressive liver disorders with parallel involvement of the brain [8,9]. TAA is metabolized to very reactive metabolites such as thioacetamide sulfine and sulfene, and these compounds can permanently bind to various macromolecules in hepatocytes [10], resulting in hepatic necrosis [11], hyperammonemia [12,13,14] and widespread oxidative stress [12,15]. Acute liver failure (ALF) and HE are associated with multiple pathological consequences, including neurological dysfunction, which contribute to mortality and are also a challenge to manage in the clinic. During HE, microglia activation and neuroinflammation occur due to dysregulated cell signaling and increased toxic metabolites in the brain [16]. There is evidence that suggests when infection occurs in an acute liver failure state, the resulting pro-inflammatory mechanisms may be amplified and could, in turn, have a major impact on blood–brain barrier (BBB) function [17]. HE can be a life-threatening complication of fulminant hepatic failure. By understanding the pathophysiology involved in the induction of this neuropsychiatric disorder, future therapeutic and/or preventive attempts could be considered [18]. Brain edema is an important complication in the course of a patient with acute liver failure and HE. Several lines of evidence suggest that increased brain water content (BWC) is seen in all forms of HE. In the pathogenesis of HE, urea and ammonia play a key role in increasing BWC. In acute liver failure, an osmotic disturbance in astrocytes, in combination with altered cerebral blood flow, leads to overt cerebral edema and intracranial hypertension [19]. Defective learning/memory ability is a feature of HE. However, the exact pathophysiological mechanisms leading to the impairment of cognitive ability in HE remain not clearly understood [20]. Treatment of HE with drugs acting on the brain targets of this syndrome is unsatisfactory [21].

Berberine (Ber), as an isoquinoline alkaloid, is mainly isolated from the Berber plant. The genus Mahonia comprises several species that contain berberine. Berberine is known for its antimicrobial activity in Asian medicine. Coptidis rhizoma (rhizomes of Coptis chinensis), another plant that contains berberine, is a famous herb very frequently used in traditional Chinese medicine for the elimination of toxins, for the treatment of “damp-heat syndromes” and to “purge fire” and “clear heat in the liver” [22,23,24]. Ber is widely used in traditional medicine and has an extensive range of biological activities including antioxidant, anti-inflammatory, anticancer, anti-hypertensive, neuroprotective and anti-hyperglycemic effects [25,26]. Despite its beneficial effects, acute and chronic toxicities have also been reported for different concentrations. Acute toxicity of Ber hydrochloride in mice indicated the following LD_50_ values: 9.0386 mg/kg (IV) and 57.6103 mg/kg (IP). Treating rats with 50 mg/kg Ber demonstrated that it has no obvious toxic effect on the kidney and liver [27]. Ber may reduce the level of reactive oxygen species (ROS) by inhibiting the N-methyl D-aspartic acid (NMDA) receptor. Since ROS are involved in the apoptotic pathway, the reduction in ROS production may be responsible for the neuroprotective effects of Ber [28]. Furthermore, it was confirmed that Ber has a hepatoprotective effect in liver ischemia/reperfusion injury and doxorubicin- and tetrachloride-induced acute hepatotoxicity [29,30,31]. Previous studies have shown that Ber increases the activity of superoxide dismutase (SOD), glutathione peroxidase (GPx) and catalase (CAT), while decreasing oxidative stress markers such as malondialdehyde (MDA), as an index of lipid peroxidation, in cell membrane activity [32] and nitric oxide (NO) levels [29,33].

In this study, an attempt has been made to determine the mechanisms involved in the effects of Ber on spatial learning and memory, BBB permeability, brain water content, hippocampal tissue inflammation and oxidative stress in TAA-induced ALF and acute HE (type A) in adult male rats.

## 2. Materials and Methods

### 2.1. Agents

The following agents were used: thioacetamide (TAA, Titrachem, Bazar Kimia, Tehran, Iran), berberine (Ber, Sigma-Aldrich Co., USA), MK-801 (Sigma-Aldrich Co., USA), DMSO (Panreac, Milan, Italy), protease inhibitory cocktail (Sigma-Aldrich Co., USA), sodium thiopental (Nesdonal, Daroopakhsh Co., Tehran, Iran), ELISA Kits for oxidative stress and interleukins assay (ZellBio GmbH, Lonsee, Germany), Evans blue (Eb, Sigma-Aldrich Co., USA).

### 2.2. Animals

One hundred fifty-six adult male Wistar rats (*n* = 156, weighing 200–250 g) were obtained from the animal care and breeding center of Ahvaz Jundishapur University of Medical Sciences (AJUMS), Ahvaz, Iran. All animals were kept in standard cages under controlled environmental conditions (temperature (22 ± 2 °C), humidity (50–55%)) and a 12 h light/dark cycle (light on at 07:00 a.m.), with free access to food chow pellets and tap water *ad libitum*. All experimental protocols were performed according to National Institute of Health (NIH) guidelines and approved by the Local Ethics Committee of the National Institute for Medical Research Development (NIMAD) (Ethic code: IR.NIMAD.REC1396.374).

### 2.3. Experimental Protocols

Following 3 days of handling, rats were divided randomly into 6 main groups with 26 rats in each.

(1) Control, rats received normal saline (2 mL/kg, i.p.) as TAA vehicle once every 48 h for 14 consecutive days. 

(2) HE, rats received TAA (200 mg/kg/2 mL normal saline, i.p.) once every 48 h for 14 consecutive days to induce the experimental rat model of acute liver failure followed by HE and 2 mL of DMSO 5%, i.p., once daily starting 24 h after the last injection of TAA.

(3–5) Treated HE groups, rats received Ber (10, 30 or 60 mg/kg/2 mL DMSO 5%, i.p., respectively) once daily starting 24 h after the last injection of TAA (18th day) until the end of behavioral tests (30th day); groups were named Ber10, Ber30 and Ber60.

(6) MK-801; rats received MK-801 (2 mg/kg/2 mL normal saline, i.p.) as a non-competitive NMDA antagonist just before TAA (once every 48 h for 14 consecutive days).

Each main group was divided into the following sub-groups:

1.Spatial memory evaluation in Morris water maze (MWM) and then brain histological evaluation (*n* = 8).2.BBB function measurement (*n* = 5). Serum biomarkers such as urea, total bilirubin concentration, liver enzyme levels, total bilirubin concentration in brain tissue.3.Brain water content measurement (*n* = 5).4.Levels of MDA, GPx, tumor necrosis factor alpha (TNF-α) and interleukin-10 (IL-10) as cytokines in hippocampal tissue (*n* = 8). The timeline and experimental protocols are shown in Figure 1.

### 2.4. Induction of Experimental Hepatic Encephalopathy (HE)

In some previous studies, fulminant hepatic failure (FHF) induced by TAA injection (300–350 mg/kg, i.p./daily for consecutive 3 days) [34,35] increased mortality, resulting in a death rate of 60%, and did not allow animals to survive for a long time. Therefore, we conducted a pilot experiment to find an appropriate dose of TAA to induce acute liver failure followed by HE, while also permitting the animals to live for at least 14 days so that spatial learning and memory testing could be performed. Finally, TAA-induced liver failure was accomplished by injections of TAA at 200 mg/kg once every 48 h for 14 days to classify both the acute liver failure and the development of HE. Twenty-four hours after the initial injection of TAA, all animals received 5% dextrose (25 mL/kg body weight, i.p.) containing 0.45% sodium chloride daily and potassium chloride (20 mEq/L) at 12 h intervals to prevent hypoglycemia, weight loss and renal failure as side effects of TAA treatment. To establish liver failure, some related biofactors such as serum urea and total bilirubin, liver enzymes including alanine aminotransferase (ALT) and aspartate aminotransferase (AST), and brain level of total bilirubin were measured. 

### 2.5. Spatial Learning and Memory Evaluation

Morris water maze (MWM) is an established test for spatial learning and memory assessment. This apparatus included a black-colored circular pool (height 60 cm, diameter 150 cm) filled with water (28 ± 1 °C) to a depth of 40 cm. The pool was divided geographically into four equal quadrants: north (N), east (E), south (S) and west (W). A black escape platform (10 cm diameter) was placed in the northeast quadrant; it was submerged 2 cm below the surface of the water and therefore was invisible. A digital camera was mounted 2 m above the maze to track the animal’s swimming path. The escape latency, swimming speed and percentage of the time spent in the target quadrant during the probe trial were measured using a video-tracking system (Ethvision software ver.7, Noldus Co., Wageningen, The Netherland). All rats underwent four trials per session during 4 consecutive days (training sessions). Animals were allowed to swim for a 60 s trial each day before the acquisition tests by placing them into the pool individually in order to familiarize them with the experimental protocol to prevent any additional stress during tests. During the training, each animal was randomly placed in one quadrant (N, E, S, or W), and the rats were allowed to freely swim for 60 s to find the hidden platform. The intertrial interval was 60 s. Twenty-four hours after the last acquisition trial (on the 5th day), a probe trial was conducted to evaluate spatial memory retrieval by removing the hidden platform. During this test, the rats swam freely for 60 s, and the time spent in the target quadrant (where the platform was placed during acquisition) was recorded as working memory retention [36,37].

### 2.6. Serum Biochemical Factor Assay

At the end of the behavioral tests, the mice were deeply and irreversibly anesthetized with an overdose of sodium thiopental (Nesdonal, 80 mg/kg, i.p.). Blood samples were collected from the heart left ventricle and were centrifuged (3000× *g*, 4 °C, 20 min) to separate serum. Serum urea and total bilirubin, brain level of bilirubin, and liver enzyme levels in serum (AST and ALT) were measured calorimetrically using commercial kits two times: the first just after the last injection of TAA (*n* = 5), and the second at the end of treatment with Ber. 

### 2.7. BBB Permeability Measurement

Blood–brain barrier (BBB) permeability was monitored by measuring extravascular Evans blue (Eb) dye in the brain and using a spectrophotometer device. In order to measure the BBB function, the rats were anesthetized, and brain vascular permeability was measured employing the injection of Eb dye via the tail vein. Briefly, animals were anesthetized with an overdose of Nesdonal; next, 20 mg/kg Eb dye 2% (1 mL/kg) was injected through the tail vein. One hour later, the thorax was opened, the descending aorta was clipped and the right atrium was cut. Then, 200–300 mL isotonic saline solution was infused into the left ventricle for 20 min to remove intravascular Eb dye. Next, the brain was immediately removed from the skull, weighed and homogenized using phosphate-buffered saline (PBS). In order to precipitate the protein, trichloroacetic acid was added. Each preparation was incubated (2–3 min at 4 °C) and was centrifuged at 3000 rpm for 20 min. In the final stage, with equal volumes, ethanol was used to dilute the supernatant, which was measured in a spectrophotometer (620 nm). The distribution of Eb dye (μg) in brain tissue (g) was calculated using the following formula: Evans blue dye in brain tissue (μg/g) = (13.24 × 20 × absorbance)/tissue weight.

Increasing vascular permeability and more severe BBB disruption were demonstrated by the increasing quantities of Eb dye found in the brain tissue compared to the control group [38].

### 2.8. Brain Water Content Measurement

A technique for cerebral edema measurement is the dry weight/wet weight method. For this purpose, rats were anesthetized irreversibly and decapitated. Their brains were carefully removed from the skulls. Immediately, the brain tissue was placed in a container (previously weighed with a digital scale), and the wet weight (WW) was measured. Then, the brain was dried in an oven at 110 °C for 24 h. After this period, the dried weight was measured again with a digital scale. The percentage of water content (% brain water content; % BWC) and brain edema (ΔH_2_O) was calculated using the following formula [39]: BWC% = [(ww − DW)/WW] × 100

### 2.9. Inflammatory Cytokine Assay

Twenty-four hours after performing all tests (31st day as shown on the timeline in Figure 1), the rats were irreversibly anesthetized with an overdose of sodium thiopental (Nesdonal 80 mg/kg, i.p.). Afterward, the animals were decapitated, and their brains were quickly removed from the skulls and rinsed with cold saline; hippocampal tissues were quickly separated on the ice, cleaned with saline and frozen at −80 °C. The samples were homogenized in a PBS solution with a proportion of 1/10 and centrifuged at −4 °C and 3000× *g* for 15 min. Then, the supernatant was used to measure the cytokines. The hippocampal tissue content of interleukin-10 (IL-10) and tumor necrosis factor alpha (TNF-α) [38] was measured using specific ELISA kits (ZellBio Gmbh, Cat. No: ZB-TNF-96A, and Cat. No: ZB-IL-10-96A, Lonsee, Baden-Wurttemberg, Germany, respectively). Briefly, the specimens were poured into wells that contained anti-IL-1β antibodies. Then, the conjugated secondary antibody was added to the medium with biotin. After the addition of streptavidin-HRP, the specimen was incubated at 37 °C for 60 min. Then, it was rinsed 5 times with saline. The chromogen was added to the medium, and 30 min later the stopping solution was added to the medium. The changed color of chromogen was read in the 450 nm optical range. Brain contents of IL-10 and TNF-α were expressed in picograms in 1 mg of protein [9,40,41].

### 2.10. Oxidative Stress Assay

An ELISA kit (ZellBio GmbH, Cat. No. ZB-MDA-96A, Lonsee, Baden-Wurttemberg, Germany) was used to measure the MDA according to the manufacturer’s guidelines [32]. The level of MDA, an index for cell membrane lipid peroxidation, was determined by thiobarbituric acid reactive substance (TBAR) assay. Results are reported as nmol of MDA per milligram of protein (nmol/mg protein) [40,42]. 

Hippocampal GPx activity was evaluated using an ELISA kit (ZellBio GmbH, Cat. No: ZB-GPx-96A, Lonsee, Baden-Wurttemberg, Germany). In this procedure, xanthine and xanthine oxidase are used to make glutathione peroxide radicals, which react with 2-(4-iodophenyl)-3-(4-nitrophenol)-5phenyltetrazolium chloride [43] to form a red Formosan color. The activity of GPx in the samples was defined by the inhibition degree of this reaction and is represented as units/mg of protein [44].

### 2.11. Histopathological Study of the Hippocampus

At the end of behavioral tests, brain hippocampi were removed from the skulls and cleaned using an ice-cold saline solution. The samples were fixed in a 10% neutral buffered formalin solution. Specimens were embedded in paraffin and sectioned (5 µm). Slides were prepared and stained with hematoxylin and eosin (H&E). The prepared tissue slides were examined under a microscope in a random order [45].

### 2.12. Statistical Analysis

The results are presented as mean ± SEM, and the data normality was checked using the Kolmogorov–Smirnov test. The data of the MWM test were analyzed using repeated measures-ANOVA followed by Tukey’s *post hoc* test. Other data were analyzed by one-way ANOVA followed by Tukey’s *post hoc* test. *p* < 0.05 was assigned as a significant difference between experimental groups. All statistical analyses were conducted using GraphPad Prism software (version 8, GraphPad Software Inc., San Diego, CA, USA).

## 3. Results

### 3.1. Spatial Learning and Memory

Figure 2A–C show spatial learning and memory in the Morris water maze for all tested groups. Escape latency (Figure 2A), swimming speed (Figure 2B) and time spent in the goal quadrant in the probe trial for working memory retrieval (Figure 2C) were measured. As indicated in Figure 2A, escape latency was significantly increased in the HE group compared to the control (*p* < 0.001), while treatment with 10, 30 and 60 mg/kg Ber reversed latency significantly (*p* < 0.01, *p* < 0.001 and *p* < 0.001, respectively). Administration of MK-801 could be effective against the negative effect of TAA in impairing spatial learning (*p* < 0.001, Figure 2A). 

Figure 2B illustrates that the swimming speed was not altered in all tested groups. Figure 2C shows the time spent in the target quadrant was significantly decreased in the HE group compared to the control (*p* < 0.001), while it increased in groups treated with different doses of Ber in dose-dependent relation with HE (*p* < 0.5 and *p* < 0.001). Administration of MK-801 could prevent memory impairment due to TAA compared to the HE group (*p* < 0.001).

### 3.2. Serum Biochemical Biomarkers

Figure 3 shows concentrations of urea (Figure 3A), total bilirubin in serum (Figure 3B), total bilirubin in brain tissue (Figure 3C) and serum levels of liver enzymes such as AST (Figure 3D) and ALT (Figure 3E). The serum urea concentration increased significantly in rats after HE induction by TAA (*p* < 0.001), while treatment with different doses of Ber reversed it significantly dose-dependently compared to HE rats (*p* < 0.05, *p* < 0.01 and *p* < 0.001, respectively). Administration of MK-801 caused a significantly lesser elevation of urea serum than that found in the HE group (*p* < 0.05). Total bilirubin concentration in serum was elevated significantly in rats after HE induction compared to the control group (*p* < 0.001). Treatment with 10, 30 and 60 mg/kg doses of Ber reversed it significantly compared to HE rats (*p* < 0.01, *p* < 0.001 and *p* < 0.001, respectively). Administration of MK-801 caused a significantly lesser elevation of total bilirubin in serum than TAA alone (HE group) (*p* < 0.001). The level of brain total bilirubin was elevated significantly in HE rats compared to the control group (*p* < 0.001). Treatment with doses of 30 and 60 mg/kg of Ber reversed it significantly compared to HE rats (*p* < 0.05 and *p* < 0.001, respectively). Administration of MK-801 caused a significantly lesser elevation of total bilirubin in the brain than TAA alone (HE group) (*p* < 0.01). 

The levels of liver enzymes AST and ALT in serum were significantly increased due to acute liver failure and HE induction by TAA compared to the control (*p* < 0.001), while treatment with Ber (30 and 60 mg/kg) reversed AST levels in serum (*p* < 0.001 and *p* < 0.001, respectively). Administration of MK-801 caused a significantly lesser elevation of serum AST than TAA alone (HE group) (*p* < 0.01).

All three doses of Ber decreased ALT significantly as a dose-dependent effect compared to HE rats (*p* < 0.05, *p* < 0.01 and *p* < 0.001, respectively). Administration of MK-801 caused a significantly lesser elevation of ALT in serum than TAA alone (HE group) (*p* < 0.05).

### 3.3. Blood–Brain Barrier Permeability

Figure 4 shows the Evans blue dye concentration in brain tissue as an index of BBB permeability alteration in different tested groups. The BBB was disrupted in the HE group, and its permeability was increased significantly compared to the control (*p* < 0.001), while treatment with different doses of Ber improved it significantly as a dose-dependent effect compared to HE rats (*p* < 0.01, *p* < 0.001 and *p* < 0.001, respectively). Administration of MK-801 significantly prevented BBB disruption due to TAA compared to the HE group (*p* < 0.001). 

### 3.4. Brain Water Content

Figure 5 illustrates the percent of water content in brain tissue (BWC) in different tested groups. The BWC was significantly and abnormally increased in the HE group compared to the control (*p* < 0.001). Treatment with 30 and 60 mg/kg doses of Ber decreased it significantly compared to HE rats (*p* < 0.05 and *p* < 0.01, respectively). Administration of MK-801 significantly prevented its increase due to TAA compared to HE (*p* < 0.001).

### 3.5. Inflammatory Cytokines in Hippocampal Tissue

Figure 6 reveals the levels of TNF-α as a pro-inflammatory cytokine (Figure 6A) and IL-10 as an anti-inflammatory cytokine (Figure 6B) in the hippocampal tissue of the different tested groups. The level of TNF-α was significantly increased after HE induction by TAA compared to the control (*p* < 0.001). Treatment with 10, 30 and 60 mg/kg Ber doses reversed its level compared to that found in HE rats (*p* < 0.05, *p* < 0.01 and *p* < 0.001, respectively). Administration of MK-801 caused a significantly lesser elevation of TNF-α in the hippocampus than TAA alone compared to the HE group (*p* < 0.001). 

The level of IL-10 in the hippocampal tissue of the HE group was decreased significantly in comparison with the control (*p* < 0.001), while treatment with Ber (30 and 60 mg/kg) significantly reversed its level compared to HE rats (*p* < 0.05 and *p* < 0.001, respectively). Administration of MK-801 could significantly prevent the decline in IL-10 in the hippocampus due to TAA compared to the HE group (*p* < 0.001). 

### 3.6. Oxidative Stress in Hippocampal Tissue

The effects of Ber on oxidative stress parameters such as MDA and the activity of GPx as an antioxidant enzyme are illustrated in Figure 7. As shown in Figure 7A, the concentration of MDA was increased significantly in the HE group compared to the control (*p* < 0.001). Treatment with doses of 10, 30 and 60 mg/kg Ber decreased the MDA concentration significantly compared to the HE group (*p* < 0.01 and *p* < 0.001, respectively). In the HE group, GPx activity decreased significantly when compared with control group (*p* < 0.001, Figure 7B). Treatment with doses of 30 and 60 mg/kg Ber significantly increased the activity of this antioxidant enzyme compared to the HE group (*p* < 0.05 and *p* < 0.01, respectively). In addition, the administration of MK-801 could significantly prevent the elevation of the MDA level and the decrement of GPx activity in the hippocampus due to TAA compared to the HE group (*p* < 0.01). 

### 3.7. Histopathology of Hippocampus

The histological findings of H&E staining showed that normal neurons with central large vesicular nuclei and peripheral distribution of Nissl granules were observed in the control group. Neurons with dystrophic changes in the form of shrunken hyperchromatic nuclei and abnormal Nissl granule distribution were found in the HE group. Treatment with Ber could reduce the number of abnormal neurons dose-dependently. Neurons with dystrophic changes were also observed in the MK-801-treated animals (Figure 8).

## 4. Discussion

In this study, TAA-induced acute liver failure that resulted in acute HE caused impairment of spatial learning and memory performances in the Morris water maze (MWM) tests. These disturbances were associated with BBB breakdown and enhanced brain water content, increased oxidative stress, neuroinflammation and neuronal loss in the hippocampus of HE rats. In contrast, treatment with berberine ameliorated all the above-mentioned alterations in TAA-induced HE rats. Previous studies have confirmed the impairment of learning and memory in experimental models of HE [46,47]. Although HE is a complex clinical condition with various effective factors, hyperammonemia seems to play a key role in the associated cognitive dysfunction [48]. Furthermore, it has been found that hyperammonemia alters the different stages of glutamatergic neurotransmission [49]. Glutamatergic neurotransmission has a key role in modulating learning and memory. So, disruption of glutamatergic neurotransmission could contribute to the impairment of learning and memory performance [49,50]. On the other hand, studies have shown that acute hepatic failure causes an increase in calcium influx via stimulation of NMDA receptors, which activate several pathways involved in free radical generation and increased ROS production in the brain [51]. Usually, in advanced stages of chronic liver disease, a remarkable increase is observed in peripheral inflammatory cytokines such as TNF-α and IL-1β, which will amplify the cascade of inflammatory responses and worsen liver function. In addition, peripheral cytokines may cross the BBB directly via passive or active transport in brain areas, thereby inducing neuroinflammation [52]. Furthermore, some studies have shown that neuroinflammation may have an important role in the development of cognitive deficits observed in animal models of HE following liver failure [53,54]. In the present study, the MWM test as a reliable paradigm of cognitive function was used to assess the potential protective effect of Ber on impaired spatial learning and memory in the HE model. Interestingly, in our findings, Ber prevented spatial memory impairment following HE induction. In agreement with our results, a large number of studies have reported that Ber treatment can improve cognitive impairment in different experimental models in rodents [55,56,57]. Additionally, it has been described that oral administration of Ber improved learning and memory deficit in Alzheimer’s disease in rats [58]. Likewise, several investigations showed that Ber decreases latency and modifies learning and memory dysfunction by improving oxidative stress and inflammatory factors [28,59]. Current findings are in line with the previous studies that demonstrated HE induced by TAA injection causes serious cognitive deficits [60,61,62].

NMDA receptor signaling is one of the main pathways in synaptic plasticity for learning and memory [63,64,65]; nevertheless, sustained activation of NMDA receptors can impair synaptic plasticity and cognition pathway in a normal situation [66,67]. In this work, we used MK-801 as an uncompetitive NMDA antagonist to establish inactivation of the NMDA receptors to prevent TAA potential for HE induction via this path, because there is a hypothesis that TAA can induce HE via ongoing activation of the NMDA pathway [68]. Current results showed TAA weakly induced HE compacts in the absence of NMDA receptors. In addition, it was reported that Ber could ameliorate lipopolysaccharide (LPS)-induced cognitive deficits through partial suppression of apoptotic cascade, neuroinflammation and oxido-nitrosative stress [69]. Moreover, it has been reported that Ber is an NMDA antagonist and is capable of protecting neuronal cells in the brain from ischemic episodes [70]. In contrast, certain studies also have confirmed that Ber reduced NMDA receptor bindings and inhibited NMDA receptor ion channel current in the brain [71,72]. 

The present study also showed HE induction causes a significant increase in brain water content (WBC) of the animals compared to the control group. Cerebral edema is a life-threatening neurological condition characterized by brain swelling due to the accumulation of excess fluid in both intracellular and extracellular spaces. Fulminant hepatic failure causes cerebral edema to develop by disrupting the BBB. However, the mechanisms by which the mediator induces brain edema in fulminant hepatic failure remain to be elucidated [73]. A previous study has demonstrated that the generation of pro-inflammatory cytokines occurs prior to the formation of brain edema [74]. In addition, it has been confirmed that liver failure induction followed by increased BBB permeability leads to vasogenic edema, subsequent ammonia excitotoxicity and the induction of edema [75]. In the current study, we observed that Ber treatment attenuated BBB disruption and reduced BWC. It was reported that Ber ameliorates infarct volume and brain edema formation and contributes to the recovery of motor function after focal cerebral ischemia by downregulation of pro-inflammatory cytokines and upregulation of anti-inflammatory cytokines [76]. Likewise, it has been shown that Ber ameliorated BBB disruption; dramatically reduced neurological deficit scores, brain water contents and tissue injury; and decreased the nuclear accumulation of NF-κB after cerebral ischemia [77]. Moreover, some investigators were reported that Ber reduces neuronal death and apoptosis, BBB permeability, and brain edema in an animal model of traumatic brain injury. Furthermore, they proposed that Ber reduces TBI-induced brain damage by limiting the production of inflammatory mediators, rather than by a direct neuroprotective effect [78].

In the current study, the HE rats showed significant increases in the serum liver enzymes (ALT, AST) as liver dysfunction markers and the concentrations of serum urea, total bilirubin and brain total bilirubin. A study demonstrated that serum liver marker levels were raised following TAA administration and HE induction. Increased serum and brain total bilirubin and decreased albumin are also commonly used as indicators of hepatocellular damage [79]. However, rats that received Ber showed ameliorated serum levels of ALT, AST, urea and total bilirubin. Additionally, it has been confirmed that Ber exerted protective effects against acute liver failure and reduced serum levels of the liver function markers induced by lead acetate [80] and CCl4 in another experimental model of HE. In this study, Ber reduced serum levels of liver function markers (ALT, AST and ALP). These findings point to the hepatoprotective and membrane-stabilizing effects of Ber [81]. 

Our finding indicated the TAA administration resulted in an increase in pro-inflammatory cytokine TNF-α levels, and there was also a significant decrease in anti-inflammatory cytokine IL-10 levels in the hippocampus. Similar to our results, it was reported that expression levels of IL-1β, IL-6, and TNF-α in the brain were significantly increased in TAA rats; moreover, the expression levels of cerebral pro-inflammatory cytokines were positively correlated with the brain water content. They suggested that inflammation was complicated in the pathogenesis of brain edema during TAA-induced HE [74]. Our assumption in the present study is that oxidative damage may probably cause neuroinflammation, which leads to cognitive deficits, through increasing the production of pro-inflammatory cytokines. However, we also found that treatment with Ber significantly decreased TNF-α and restored IL-10 levels in the hippocampus compared with the untreated HE group. Consistent with our results, it was reported that the neuroprotective effect of Ber against learning and memory deficits in traumatic brain injury occurs through the suppression of inflammation, oxidative stress and apoptosis [55]. A growing body of evidence supports the potent anti-inflammatory effect of Ber in experimental rats [82,83]. Other researchers have also suggested that Ber is a potential agent for the treatment of liver fibrosis. It ameliorated hepatic fibrogenesis through different mechanisms, including reducing the pro-inflammatory cytokine release, strong antioxidative effect, stimulating the production of anti-inflammatory cytokine IL-10 and preventing apoptosis [84]. 

Hence, we have examined this hypothesis by assessing whether the blockade of NMDA receptors with MK-801 can prevent alterations in serum biomarkers, hippocampal antioxidant enzymes and anti-inflammatory cytokines. As shown in Figure 3, the blockade of NMDA receptors by MK-801 prevented the increase in serum urea, serum and brain total bilirubin concentrations in the HE rats. Current results reveal that the Evans blue (Eb) content in the brain tissue of the group treated with MK-801 was significantly lower than that in the HE group. Interestingly, a significant decrease in TNF-α level and a restoration of the IL-10 level were shown in the MK-801 group compared to the HE group. Moreover, injection of MK-801 also prevents the TAA-induced decrease in the GPx activity and the increase in MDA level in hippocampal tissue. These changes support the idea that the effects of hyperammonemia on oxidative stress are mediated by the hyperactivation of the NMDA receptor. These findings are in line with those of other studies [68].

On the other hand, it is also well known that hyperammonemia affects different types of glutamatergic receptors and leads to the hyperactivation of NMDA receptors, which are responsible for events such as mitochondrial dysfunction and subsequently increased production of ROS, as well as hyperactivation of calcium-dependent enzymes that cause neuronal damage. Several studies have reported that the NMDA receptor antagonist MK-801 reduces the production and release of glutamate and prevents ammonia-induced superoxide radical formation [85,86]. Experimental evidence indicates a significant increase in oxidative stress in the brains of TAA-induced HE animals [87,88]. Previously, a study reported that hyperammonemia leads to a decrease in the activities of GPx, catalase and SOD and increases the formation of superoxide in the brains of rats. In addition, it has been shown that ammonia-induced inhibition of antioxidant enzymes is mediated through the activation of NMDA receptors [89]. Additionally, in the current study, data showed a significantly increased level of MDA and decreased level of GPx in the hippocampal tissue of HE rats, which may have resulted from elevated oxidative stress. An earlier study also indicated a significant reduction in antioxidant enzyme activity and an increase in protein oxidation and lipid peroxidation produced in different parts of the brain tissue following TAA-induced HE in rats [90,91]. It was confirmed that pathologically induced oxidative stress could cause brain dysfunction and result in impaired spatial learning and memory. Additionally, some pieces of evidence support the concept of ROS and their involvement in the oxidative pathway of memory impairment [92,93]. Our findings demonstrated that Ber reduces lipid peroxidation but increases the activity of GPx, an antioxidant enzyme, in HE rats. Certain studies have shown the protective effect of Ber in hepatotoxicity through the prevention of the lipid peroxidation process and the increases in GSH levels and GPx activity in liver tissue [94]. Several studies have demonstrated the hepatoprotective effect of Ber in different pathological conditions. It has been shown that Ber is able to prevent mercury-induced liver failure in rats by preventing inflammation, apoptosis and oxidative stress [95]. In addition, the antioxidant effect of Ber is also responsible for its hepatoprotective property in lead-intoxicated mice [96]. Treatment with Ber significantly reduced the levels of nitrite and lipid peroxidation products and increased SOD activity in diabetic rats [97]. A researcher was reported that restoration of the brain tissue level of SOD and reduction in MDA contents have been proposed as responsible for the neuroprotective effect of Ber against chronic brain injury induced by aluminum trichloride in rats, therefore ameliorating cognitive dysfunction and hippocampal injury [98]. In addition, previous studies show that Ber plays an important role as an indirect antioxidant and increases the activities of antioxidant enzymes [29,95]. 

Furthermore, in the current work, histological verification of hippocampal tissue in all tested groups with H&E staining was performed to assess the histopathology of the hippocampal CA1 region. The histopathological results have shown that TAA administration caused a significant increase in the density of damaged neurons in the hippocampal CA1 region compared to the control (black arrows in Figure 8). Interestingly, treatment of HE rats with different doses of Ber caused a significant decrease in damaged neurons in the hippocampal CA1 region. Our findings are in agreement with previous studies that suggest Ber provides protection in an aluminum-induced rat model of neurodegenerative disease [99] and in the experimental model of vascular dementia [99] (Figure 9). It should be noted here that there were some limitations in the current work in performing some techniques in order to determine the expression of Bcl-2 and Bax as apoptotic and anti-apoptotic proteins in the hippocampal tissue of tested groups. 

Besides the beneficial effects of Ber in rats with acute liver damage, some studies have also shown that Ber has adverse toxic effects. It has been reported that intraperitoneal injection of Ber (10 and 20 mg/kg/day, for 1 week) decreases bilirubin protein binding in adult rats. Clinically, it should be noticed that this substance could increase the risk of kernicterus in risky patients [100]. A researcher investigated the immunotoxic effects of Ber (5 and 10 mg/kg/day, i.p., for 14 days) in BALB/c mice. Administration of Ber (10 mg/kg) decreased spleen weight; blood cell count, including the number of leukocytes, neutrophils and lymphocytes; and diminished generation/differentiation of B and T cells. Ber at 10 mg/kg suppressed both cellular and humoral immune functions; at 5 mg/kg, it only influenced the proliferation of lymphocytes and the delayed-type hypersensitivity response [101]. Furthermore, in the current study, we also observed that the administration of Ber (100 mg/kg, i.p.) caused 100% mortality in rats after the second injection.

In conclusion, our findings provide some support for the beneficial effects of Ber on spatial learning and memory impairment via amelioration the brain inflammation and oxidative stress in an animal model of HE. So, Ber in appropriate doses may act as a potential clinical agent, which can ameliorate some HE complications. Nevertheless, we could not evaluate some possible signaling pathways involved. The precise mechanisms of the neuroprotective effect of Ber in an animal model of HE remain to be elucidated.

## Figures and Tables

**Figure 1 brainsci-13-00944-f001:**
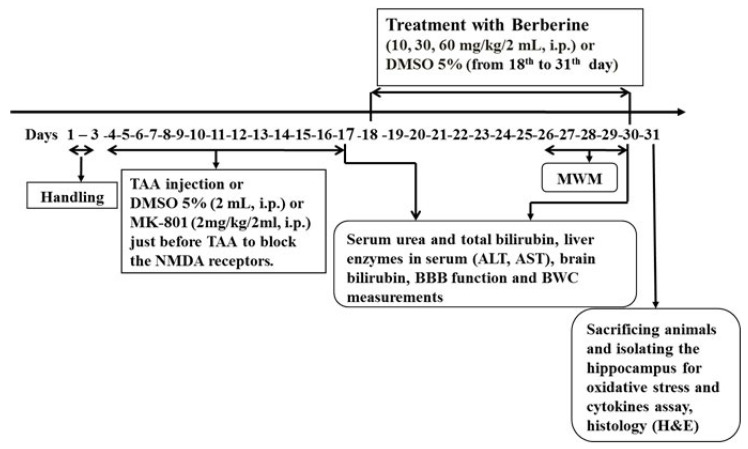
The timeline and experimental design. Treatment of HE rats with berberine started on the 18th day (24 h after last TAA administration) and continued until the end of behavioral tests.

**Figure 2 brainsci-13-00944-f002:**
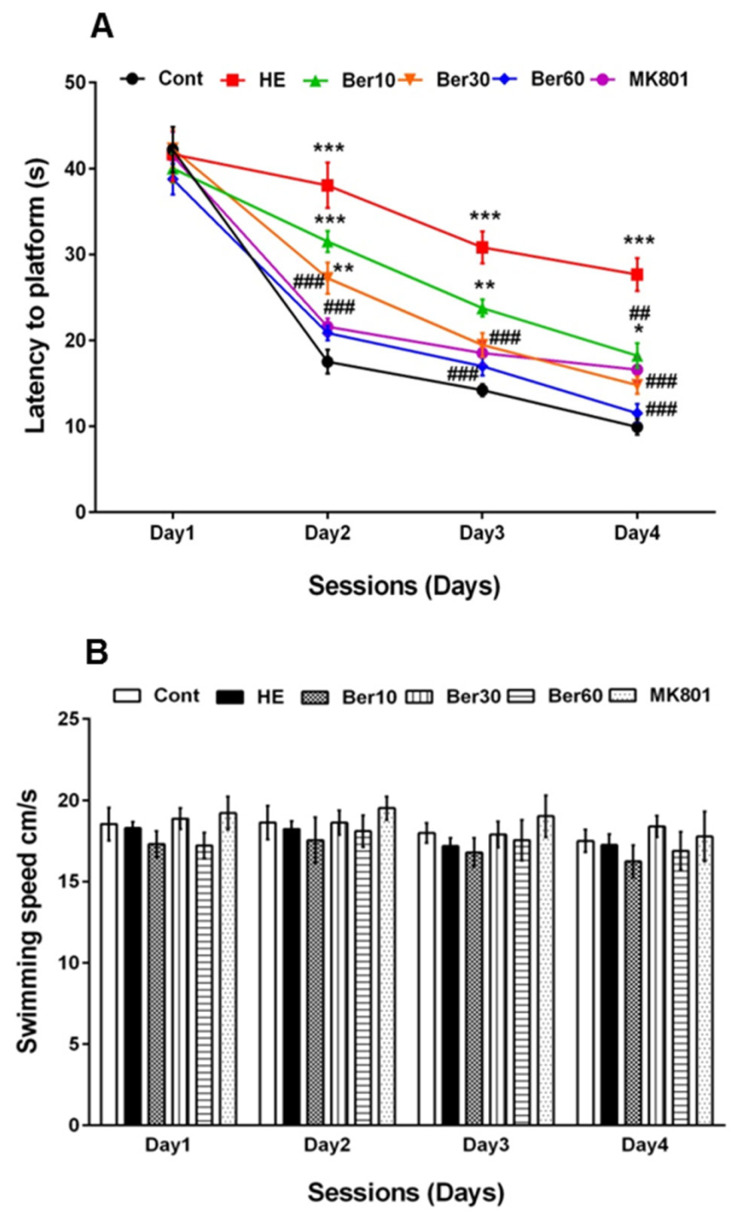
Spatial learning and memory of all tested groups evaluated in the Morris water maze. Data are represented as mean ± SEM. (**A**) Latency (s) in finding and jumping on hidden platform. (**B**) Swimming speed. (**C**) Probe trial to measure the time spent in goal quarter as working memory retrieval. Data analyzed by repeated measure ANOVA followed by Tukey’s post hoc test (*n* = 8). ** *p* < 0.01 and *** *p* < 0.001 vs. control group, ## *p* < 0.001, ### *p* < 0.001 vs. HE group. HE: hepatic encephalopathy, Ber: berberine. * vs. Cont, # vs. HE.

**Figure 3 brainsci-13-00944-f003:**
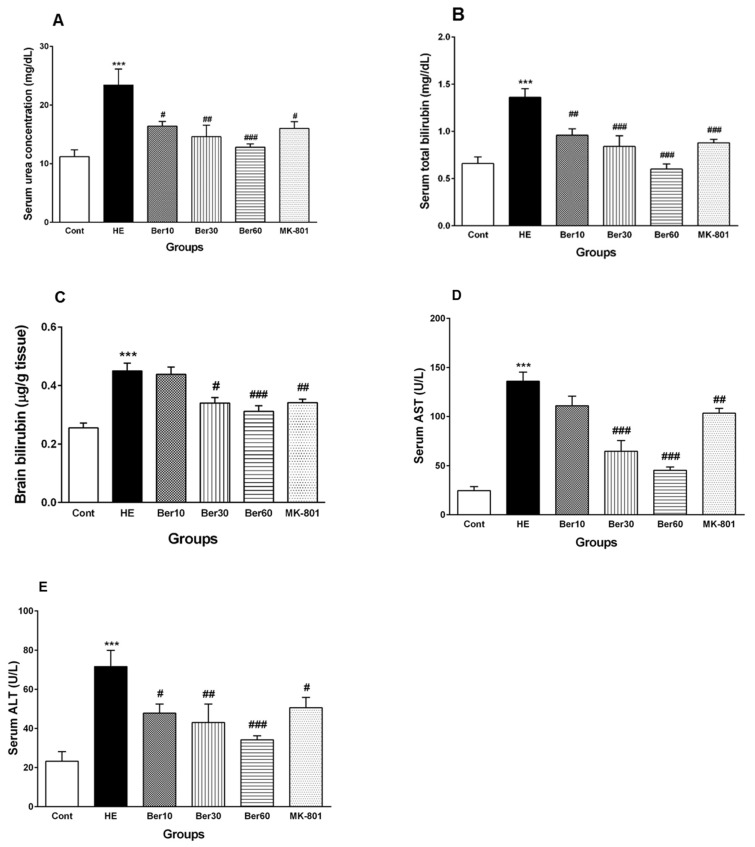
Biochemical biomarkers, including (**A**) serum urea, (**B**) serum total bilirubin concentrations, (**C**) total bilirubin concentration in brain tissue, (**D**) AST and (**E**) ALT liver enzyme levels in serum. Data are represented as mean ± SEM. Data analyzed by one-way ANOVA followed by Tukey’s post hoc test (*n* = 8). *** *p* < 0.001 vs. control group, # *p* < 0.05, ## *p* < 0.01 and ### *p* < 0.001 vs. HE group. HE: hepatic encephalopathy, Ber: berberine, AST; aspartate aminotransferase, ALT: alanine aminotransferase.

**Figure 4 brainsci-13-00944-f004:**
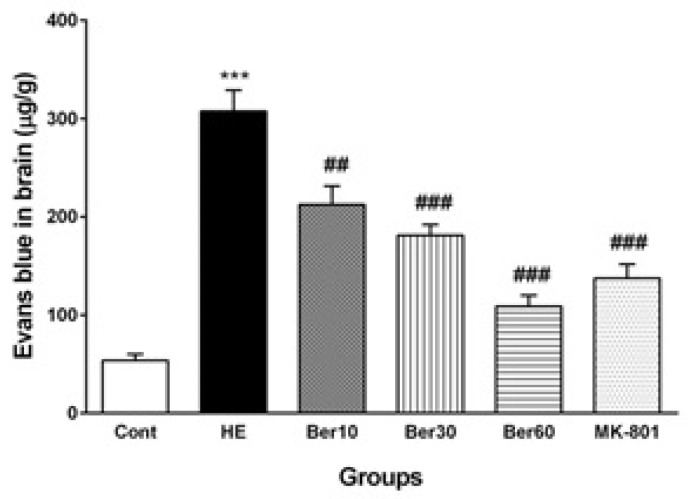
The levels of Evans blue dye in brain tissue (*n* = 5). Data are represented as mean ± SEM and were analyzed by one-way ANOVA followed by Tukey’s post hoc test. *** *p* < 0.001 vs. control group, ## *p* < 0.01 and ### *p* < 0.001 vs. HE group. HE: hepatic encephalopathy, Eb: Evans blue dye, Ber: berberine.

**Figure 5 brainsci-13-00944-f005:**
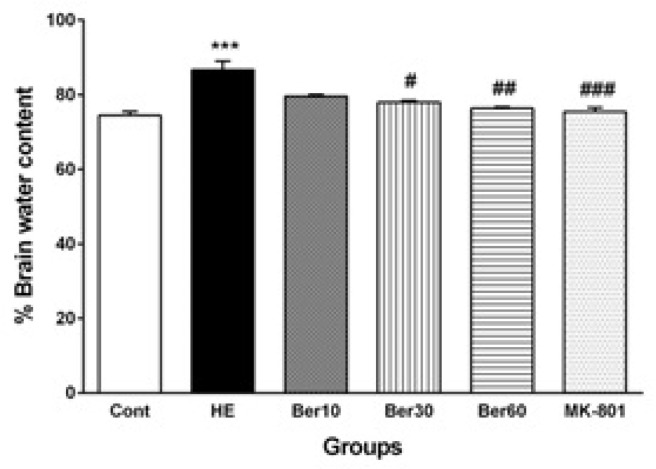
The levels of brain water content (*n* = 5). Data are represented as mean ± SEM and were analyzed by one-way ANOVA followed by Tukey’s post hoc test. *** *p* < 0.001 vs. control group, # *p* < 0.05, ## *p* < 0.01 and ### *p* < 0.001 vs. HE group. HE: hepatic encephalopathy, Ber: berberine.

**Figure 6 brainsci-13-00944-f006:**
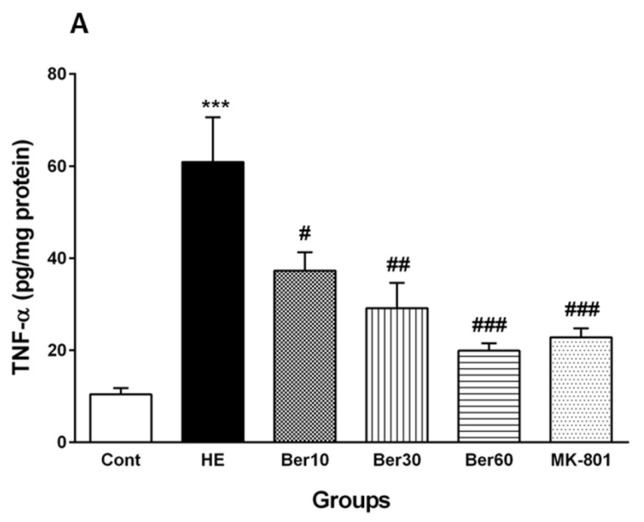
Hippocampal levels of tumor necrosis factor alpha (TNF-α) (**A**) and interleukin-10 (IL-10) (**B**) in all experimental groups. Data are represented as mean ± SEM and were analyzed by one-way ANOVA followed by Tukey’s post hoc test (*n* = 8). *** *p* < 0.001 vs. control group, # *p* < 0.05, ## *p* < 0.01 and ### *p* < 0.001 vs. HE group. HE: hepatic encephalopathy, Ber: berberine.

**Figure 7 brainsci-13-00944-f007:**
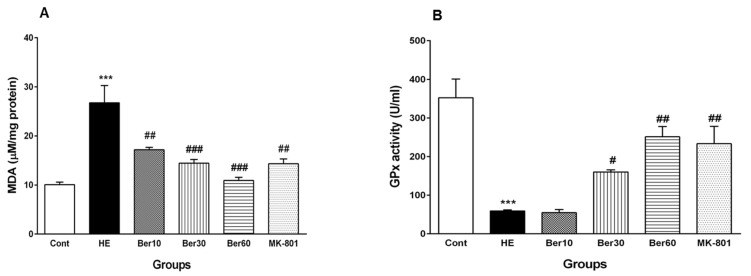
Oxidative stress in hippocampal tissue of all experimental groups. Data are represented as mean ± SEM. Malondialdehyde (MDA) as a lipid peroxidation index (**A**) and activity of glutathione peroxidase (GPx), an antioxidant enzyme (**B**) in hippocampal tissue. Data analyzed by one-way ANOVA followed by Tukey’s post hoc test (*n* = 8). *** *p* < 0.001 vs. control group, # *p* < 0.05, ## *p* < 0.01 and ### *p* < 0.001 vs. HE group. HE: hepatic encephalopathy, Ber: berberine.

**Figure 8 brainsci-13-00944-f008:**
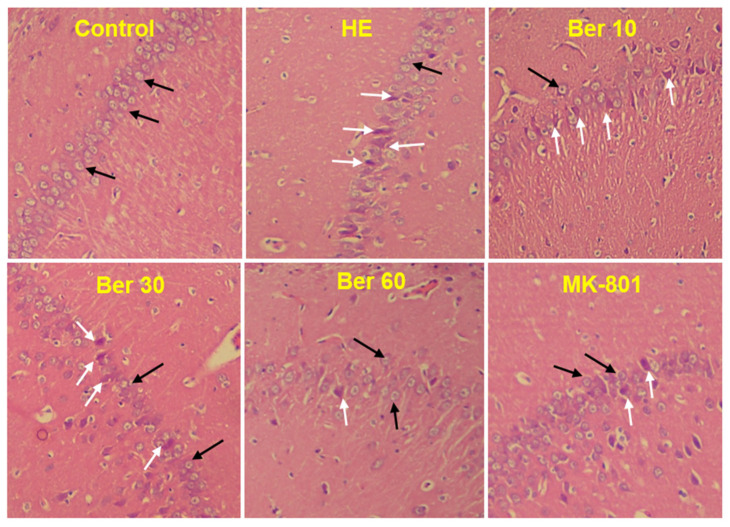
Photomicrograph of brain hippocampal CA1 region of different groups. Black arrows indicate normal neurons with central large vesicular nuclei and peripheral distribution of Nissl granules. White arrows indicate neurons with dystrophic changes in the form of shrunken hyperchromatic nuclei and abnormal Nissl granule distribution. Magnification: 250×. HE: hepatic encephalopathy, Ber: berberine.

**Figure 9 brainsci-13-00944-f009:**
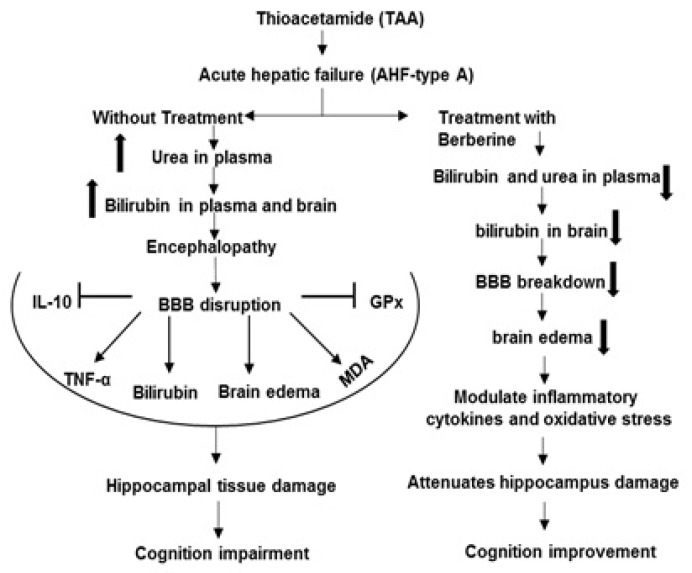
The mechanisms of beneficial effects of Ber on complications of hepatic encephalopathy induced by TAA. Symbols: Inhibition 
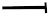
, Stimulation 
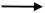
, Increase 
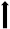
, Decrease 
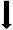
.

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
