# Peer review of "Effect of Berberine against Cognitive Deficits in Rat Model of Thioacetamide-Induced Liver Cirrhosis and Hepatic Encephalopathy (Behavioral, Biochemical, Molecular and Histological Evaluations)"

_brainsci, 2023, doi:10.3390/brainsci13060944_

Round 1
Reviewer 1 Report
The authors investigated the effect of berberine on TAA-induced liver cirrhosis with accompanying hepatic encephalopathy. The obtained results are scientifically significant with potential clinical use. I suggest a few changes to improve the manuscript.
1. In the abstract instead of "6) MK-801+TAA; received MK-801" should be stated "6) dizolcipine (MK-801)+TAA; received MK-801”. In this way, both names of this uncompetitive antagonist of NMDA receptor are immediately introduced. In the rest of text there is no need to use both names anymore, it is enough to use only MK801 everywhere, including in Figure legends.
2. The text “(an uncompetitive NMDA receptor antagonist) could be removed from Figure legends and Results, since MK801 pharmacological use is already mentioned in 2.3. Experimental protocol part of manuscript.
3. It seems that there are some errors in experimental design.
A) First, it is not clear what is TAA vehicle and what is the role of DMSO in control group (2.3. Experimental protocols part of manuscript). To be more precise, DMSO is Ber vehicle given once daily from 24 h after the last injection of TAA (18th day) until the end of behavioral tests (30th day). Therefore, control group, beside the i.p. saline as the TAA vehicle once every 48 h for consecutive 14 days, should received DMSO once daily from 24 h after the last injection of TAA (18th day) until the end of behavioral tests (30th day), in a way similar to HE group of animals. Maybe is the type error? Please explain.
B) There is a statement in the same subchapter that HE rats received TAA (200 mg/kg/2 ml normal saline, i.p.) + 2ml of DMSO 5% once daily from 24 h after the last injection of TAA. It should be added i.p, because it is the way of DMSO application. So the whole sentence could be “HE; rats received TAA (200 mg/kg/2 ml normal saline, i.p.) once every 48 h for consecutive 14 days to induce the experimental rat model of acute liver failure followed by HE and 2 ml of DMSO 5% i.p. once daily from 24 h after the last injection of TAA.
C) In the same subchapter there is sentence “MK-801; rats received dizolcipine (MK-801, 2 mg/kg/2 ml, i.p.) as a non-competitive NMDA antagonist just before TAA (once every 48 h for 14 consecutive days). What is the vehicle for the MK801?
4. The sample size (n=5 in case of BBB function measurement and brain water content measurement) is statistically not large enough - the minimal sample size should be at least 6. Please justify or cite similar case from literature.
Author Response
Response to Reviewers' comments
May 9, 2023 
Dear Editor,
It is with excitement that I resubmit to you a revised version of manuscript entitled "Effect of Berberine against cognitive deficits in rat model of thioacetamide-induced
liver cirrhosis and hepatic encephalopathy (behavioral, biochemical, molecular and
histological evaluations)" for the journal of Brain Sciences (ISSN 2076-3425). Thank you for giving me the opportunity to revise and resubmit this manuscript. We would like to express our appreciation for your comments. As you will see below, we have revised and improved the paper as a result of your valuable feedback. In the revised manuscript, you can find all of changes made according to your comments as underlined sentences.
Sincerely yours
* Corresponding author: Alireza Sarkaki, Prof. of Neurophysiology, Persian Gulf Physiology Research Center, Medical Basic Sciences Research Institute, Ahvaz Jundishapur University of Medical Sciences, Ahvaz, Iran.
sarkaki_a@ajums.ac.ir and sarkaki145@gmail.com
Reviewer #1:
- In the abstract instead of "6) MK-801+TAA; received MK-801" should be stated "6) dizolcipine (MK-801)+TAA; received MK-801”. In this way, both names of this uncompetitive antagonist of NMDA receptor are immediately introduced. In the rest of text there is no need to use both names anymore, it is enough to use only MK-801 everywhere, including in Figure legends.
- Many thanks for your precise suggestion. According to your comment, these sentences were revised.
- The text “(an uncompetitive NMDA receptor antagonist) could be removed from Figure legends and Results, since MK801 pharmacological use is already mentioned in 2.3. Experimental protocol part of manuscript.
- With thanks. Duplicate sentences were removed.
- It seems that there are some errors in experimental design.
- A) First, it is not clear what is TAA vehicle and what is the role of DMSO in control group (2.3. Experimental protocols part of manuscript). To be more precise, DMSO is Ber vehicle given once daily from 24 h after the last injection of TAA (18th day) until the end of behavioral tests (30th day). Therefore, control group, beside the i.p. saline as the TAA vehicle once every 48 h for consecutive 14 days, should receive DMSO once daily from 24 h after the last injection of TAA (18th day) until the end of behavioral tests (30th day), in a way similar to HE group of animals. Maybe is the type error? Please explain.
- You are right. Control group received normal saline as TAA vehicle. The mistake in the text (2.3. Experimental protocols) was corrected.
- B) There is a statement in the same subchapter that HE rats received TAA (200 mg/kg/2 ml normal saline, i.p.) + 2ml of DMSO 5% once daily from 24 h after the last injection of TAA. It should be added i.p, because it is the way of DMSO application. So the whole sentence could be “HE; rats received TAA (200 mg/kg/2 ml normal saline, i.p.) once every 48 h for consecutive 14 days to induce the experimental rat model of acute liver failure followed by HE and 2 ml of DMSO 5% i.p. once daily from 24 h after the last injection of TAA.
- With thanks. As suggested by the reviewer, i.p. was added in the text and the phrase also revised.
- C) In the same subchapter there is sentence “MK-801; rats received dizolcipine (MK-801, 2 mg/kg/2 ml, i.p.) as a non-competitive NMDA antagonist just before TAA (once every 48 h for 14 consecutive days). What is the vehicle for the MK801?
- Ok. The vehicle of MK801 was normal saline.
- The sample size (n=5 in case of BBB function measurement and brain water content measurement) is statistically not large enough - the minimal sample size should be at least 6. Please justify or cite similar case from literature.
Thanks you very much for your attention. The results of BBB function and brain water content tests were enough statistically. We used same sample size (n=5 in each test) according to statistics advisor for same tests in our previous research and we reported their results in following published papers (1).
- Somayeh Hajipour, Yaghoob Farbood, Mohammad Kazem Gharib-Naseri, et.al. Exposure to ambient dusty particulate matter impairs spatial memory and hippocampal LTP by increasing brain inflammation and oxidative stress in rats. Life sci. (242), 2020, 117210

Reviewer 2 Report
The manuscript is well-conceived and written but some drawbacks must be addressed.
Carefully review the manuscript and correct underlined.
Minor Reviews
The title can read Effect of Berberine against cognitive deficits in a rat model of thioacetamide-induced liver Cirrhosis and hepatic encephalopathy.
1. Abstract :
Background: Liver cirrhosis (LC) is one of the chronic liver diseases with high disability and 16
work on the effect of berberine (Ber) on spatial learning and memory in a rat model of LC induced by thioacetamide 18
memory, oxidative stress, cytokines in the hippocampus, and histopathology were evaluated. Results: 23
Conclusion: Results showed that Ber improves spatial memory in TAA-induced LC by improving the BBB 28
2. Introduction 34
· Liver disorders are very frequent and widely distributed in the world. Between 35
· them hepatic fibrosis represents the response of the liver to different chronic insults, and 36
· is accompanied by significant morbidity and mortality [1]. In several investigational 37
· titis and cirrhosis. Indeed, several lines of investigations have shown the advantageous 43
· toxicity for its different concentrations has also been reported. Acute toxicity of Ber hydro- 107
· inhibiting N-methyl D-aspartic (NMDA)-receptor. Since ROS contributes in apoptosis 111
· fects of Ber [31]. Furthermore, it was confirmed that Ber has a hepatoprotective effect in the 113
3. Material and Methods
· ad libitum. All experimental protocols were done according to the National Institute of Health 136
· 2.4. Induction of experimental hepatic encephalopathy (HE) 167
· In some previous studies, fulminant hepatic failure (FHF) was induced by in- 168
· 3 days) [37, 38] caused a high rate of mortality (>60%) and do not allow the animals to live 170
· chloride and potassium chloride (20 meq/l) at 12 h intervals to prevent hypoglycemia, 178
· weight loss and renal failure as side effects of treatment with TTA. To establish liver failure 179
· some related biofactors such as serum urea and total bilirubin, liver enzymes such as ala- 180
· nine aminotransferase (ALT), aspartate aminotransferase (AST), and brain level of total bilirubin were measured. 182
· ically into four equal quadrants including North (N), East (E), South (S), and West (W). A 187
· submerged 2 cm below the surface of the water and therefore was invisible. A digital camera 189
· was mounted 2 m above the maze to track the animal’s swimming path. The escape 190
· latency, swimming speed and percentage of the time spent in the target quadrant during 191
· probe trials were measured by a video-tracking system (Ethvision software ver.7, Noldus 192
· Co., Netherlands). All rats received four trials per session during 4 consecutive days (train- 193
· ing sessions). Animals were allowed to swim for 60 seconds trial a day before the 194
· acquisition tests by falling them into the pool individually in order to familiarize them to experi- 195
· ment protocol to abolish any additional stress during tests. During the experiment each ani- 196
· (60 s) and stay for 30 s. The inter-trial interval was 60 seconds. Twenty-four hours after the 199
· last acquisition trial (on the 5th day), a probe trial was conducted to evaluate spatial memory 200
· Blood-brain barrier (BBB) permeability was monitored by measuring extravascular 213
· Evans blue (Eb) dye in the brain and using a spectrophotometer device. In order to measure 214
· tized with an overdose of Nesdonal; next, 20 mg/kg Eb dye 2% (1 ml/kg) was injected through 217
· the tail vein. One hour later, the thorax was opened, descending aorta was clipped, and right 218
· the atrium was cut. Then, 200–300 ml isotonic saline solution was infused into the left ventricle 219
· 228
· Increasing vascular permeability and more severe BBB disruption were demonstrated- 229
· Their brains were carefully removed from the skull. A technique for 234
· weight (WW) was measured. After the brain was placed in an oven for 24 h at 110 °C dry 237
· (Nesdonal 80 mg/kg, ip), afterward, the animals were decapitated and their brains were quickly 246
· removed from the skull and rinsed with cold saline, hippocampi tissues were quickly sepa- 247
· rinsed 5 times with saline. The chromogen was added to the medium, and 30 min later the 256
· in the 450 nm optical range. Brain contents of IL-10 and TNF-α were expressed in 258
tal (Nesdonal 80 mg/kg, i.p.) and cleaned with an ice-cold saline solution. The samples were 277
pared tissue slides were examined under a microscope in random order. At least six 281
Kolmogorov–Smirnov test. The data of the MWM test were analyzed using repeated 286
4. Results 291
3.1. Spatial Learning and Memory 292
· Figure 2A-C shows spatial learning and memory in the Morris water maze for all tested 293
· to impair spatial learning (p<0.001, Fig. 2A). 300
· Figure 2B illustrates that the swimming speed was not altered in all tested groups. 301
· 801 could prevent memory impairment due to TAA compared to HE group (p<0.001). 305
3.7. Histopathology of hippocampus
· Scale bar should be added to the micrographs.
· H&E stains cannot be used to identify Nissl granules, the standard stain is Cresyl violet.
· Some of the neurons identified with dystrophic changes in the micrographs of HE, BER 10, BER 30, and BER 60 are actually pyramidal cells. In the CA regions, there are granule (round) and pyramidal (triangular) cells.
· Therefore, the histology results are not accurately interpreted.
· in brain tissue (C), liver enzyme levels in serum such as AST (D) and ALT (E). The con- 318
· (p<0.001). The level of total bilirubin in the brain was elevated significantly in rats after HE 328
· Administration of MK-801 caused lesser elevation of total bilirubin in the brain than TAA 331
· Figure 6 reveals the levels of TNF-α as a pro-inflammatory cytokine (A) and IL-10 as 376
· an anti-inflammatory cytokine (B) in hippocampal tissue of different tested groups. The 377
· MK-801, could prevent the decline of IL-10 in the hippocampus due to TAA significantly related 386
5. Discussion 432
· cognitive impairments following HE induction. In agreement with our results, a large 458
· number of studies have reported that Bertreatment can improve cognitive impairment 459
· and modifies learning and memory dysfunction by improving oxidative stress and 463
· vation of the NMDA pathway [71]. Current results showed TAA weakly induced HE com- 472
· memory which was impaired following HE. Moreover, it has been reported that Ber exhibits 478
· matory cytokines occur prior to the formation of brain edema [77]. In addition, it has 489
· bility, and brain edema in an animal model of traumatic brain injury. Furthermore, they 500
· In current study, HE rats showed significant increases in the serum liver enzymes 503
· (ALT, AST) as liver dysfunction markers, the concentration of serum urea, total bilirubin, and 504
· [82]. However, rats that received Ber showed ameliorated serum levels of ALT, AST, urea, total 508
· expressions of IL-1β, IL-6, and TNF- α in the brain significantly increased in TAA rats; more- 517
· with our results, Wang and Zhang 2018 reported that neuro-protective effect of Ber 525
· inflammation, oxidative stress and apoptosis [58]. Growing evidence support the potent 527
· hepatic fibrogenesis through different mechanisms including; reducing the pro-inflamma- 530
· Hence, we have examined this hypothesis by assessing whether the blockade of NMDA 533
· receptors with dizocilpine (MK-801) can prevent alterations in serum biomarkers, hippo- 534
· campal antioxidant enzymes and anti-inflammatory cytokines. As shown in figure 3, 535
· blockade of NMDA receptors by MK-801 prevents the increase in serum urea, serum and 536
· (Eb) content in the brain tissue of the group treated with MK-801 was significantly lower than 538
· the HE group. Interestingly, a significant decrease in TNF-α level and restoration of IL- 539
· 10 level showed in the MK-801 group compared to the HE group. Moreover, the injection of 540
· of hyperammonia on oxidative stress are mediated by hyperactivation of NMDA receptors. 543
· These findings are in line with other studies [71]. 544
· reduces the production and release of glutamate and prevents ammonia-induced superox- 550
· may have resulted from elevated oxidative stress. An earlier research also indicated 558
· that a significant reduction in antioxidant enzymes activity, and an increase in protein 559
· oxidation and lipid peroxidation produced in different parts of the brain tissue following 560
· TAA-induced HE in rats [93, 94]. It was confirmed that pathologically induced oxidative 561
· stress could cause brain dysfunction and result in impaired spatial learning and memory. 562
· Additionally, some evidence supports the concept of ROS and their involvement in the 563
· nitive dysfunction and hippocampal injury [101]. In addition, previous studies show 577
· that Ber plays an important role as an indirect antioxidant and increases the activities of 578
· antioxidant enzymes [32, 98]. 579
· tested groups with H&E staining was done to assess the histopathology of hippocampal CA1 581
· region. Our findings are in agreement with previous studies that suggest Ber provides 586
· there were some limitations in the current work to performing some techniques in order to 589
· Besides the beneficial effects of Ber in rats with acute liver damage, some studies have 595
· persensitivity response [105]. Furthermore, in the current study, we also observed that the ad- 605
· ministration of Ber (100 mg/kg, i.p.) caused a 100% mortality in rats after the second injection. 606
· In conclusion, our findings provide some support for the beneficial effects of Ber on 607
6. References
References should be reviewed for uniformity, take note of ref 49, 87, and 104.
Major Revision
1. The authors should thoroughly review the manuscript for grammatical, spelling, and punctuation errors.
2. The paragraph covered by lines 58-64 is unclear and should be rewritten.
3. “At the end of behavioral tests brain hippocampi of all the animals were removed from 275 the skull under deeply and irreversible anesthetized with an overdose of sodium thiopen- 276 tal (Nesdonal 80 mg/kg, i.p.)”-
How did you identify the gross hippocampi and CA1 region?
4. Morris Water Maze tested for Spatial learning and memory and not cognition, so take note of this in your discussion and conclusion.
5. “In contrast, treatment with berberine reversed all the above-men- 437 tioned alterations in TAA-induced HE rats. Previous studies have confirmed the impair- 438”
The term reversed should be changed to ameliorated
6. The authors stated that they performed neuronal cell count both in the method and discussion sections, but there are any results on cell count. The authors should either add such results or expunge the associated sections from the manuscript.
The quality of English is poor and should be improved.
Author Response
Response to Reviewers' comments
May 9, 2023 
Dear Editor,
It is with excitement that I resubmit to you a revised version of manuscript entitled "Effect of Berberine against cognitive deficits in rat model of thioacetamide-induced
liver cirrhosis and hepatic encephalopathy (behavioral, biochemical, molecular and
histological evaluations)" for the journal of Brain Sciences (ISSN 2076-3425). Thank you for giving me the opportunity to revise and resubmit this manuscript. We would like to express our appreciation for your comments. As you will see below, we have revised and improved the paper as a result of your valuable feedback. In the revised manuscript, you can find all of changes made according to your comments as underlined sentences.
Sincerely yours
* Corresponding author: Alireza Sarkaki, Prof. of Neurophysiology, Persian Gulf Physiology Research Center, Medical Basic Sciences Research Institute, Ahvaz Jundishapur University of Medical Sciences, Ahvaz, Iran.
sarkaki_a@ajums.ac.ir and sarkaki145@gmail.com
Reviewer #2:
- The authors should thoroughly review the manuscript for grammatical, spelling, and punctuation errors.
Thank you very much. According the reviewer’s suggestion the manuscript revised by an expert and native in English language, and improved to the best of my ability.
- The paragraph covered by lines 58-64 is unclear and should be rewritten.
- Thank the honor reviewer for precise attention. This paragraph was revised.
- “At the end of behavioral tests brain hippocampi of all the animals were removed from 275 the skull under deeply and irreversible anesthetized with an overdose of sodium thiopen- 276 tal (Nesdonal 80 mg/kg, i.p.)”-
How did you identify the gross hippocampi and CA1 region?
- Ok, It was identified and diagnosed based on histological studies, preparation the hippocampus tissue slices, H & E stain and determinate the CA1 region of hippocampus under microscope with proper magnification with help of a histologist advisor.
- Morris Water Maze tested for Spatial learning and memory and not cognition, so take note of this in your discussion and conclusion.
- Many thanks for your valuable comment. The discussion and conclusion was revised.
- “In contrast, treatment with berberine reversed all the above-men- 437 tioned alterations in TAA-induced HE rats. Previous studies have confirmed the impair- 438” The term reversed should be changed to ameliorated.
- Ok, thank you. The term reversed was replaced by the ameliorated.
- The authors stated that they performed neuronal cell count both in the method and discussion sections, but there are any results on cell count. The authors should either add such results or expunge the associated sections from the manuscript.
- Thank the honor reviewer for nice comment. This section removed from the manuscript.
